# Effect of Gliadin Stimulation on HLA-DQ2.5 Gene Expression in Macrophages from Adult Celiac Disease Patients

**DOI:** 10.3390/biomedicines10010063

**Published:** 2021-12-29

**Authors:** Federica Farina, Laura Pisapia, Mariavittoria Laezza, Gloria Serena, Antonio Rispo, Simona Ricciolino, Carmen Gianfrani, Alessio Fasano, Giovanna Del Pozzo

**Affiliations:** 1Institute of Genetics and Biophysics “A. Buzzati-Traverso”, Italian National Council of Research (CNR), 80131 Naples, Italy; federica.farina@igb.cnr.it (F.F.); laura.pisapia@igb.cnr.it (L.P.); mariavittoria.laezza@igb.cnr.it (M.L.); 2Division of Pediatric Gastroenterology and Nutrition, Center for Celiac Research, Mucosal Immunology and Biology Research Center, Harvard Medical School, Massachusetts General Hospital, Boston, MA 02114, USA; gloria.serena@libero.it (G.S.); afasano@mgh.harvard.edu (A.F.); 3Gastroenterology, Department of Clinical Medicine and Surgery, School of Medicine Federico II of Naples, 80131 Naples, Italy; antonio.rispo2@unina.it (A.R.); sricciolino@aslavellino.it (S.R.); 4Institute of Biochemistry and Cell Biology, Italian National Council of Research (CNR), 80131 Naples, Italy; carmen.gianfrani@ibbc.cnr.it

**Keywords:** autoimmunity, celiac disease, HLA gene expression, macrophages

## Abstract

Macrophages play an important role in the pathogenesis of celiac disease (CD) because they are involved in both inflammatory reaction and antigen presentation. We analyzed the expression of CD-associated HLA-DQ2.5 risk alleles on macrophages isolated by two cohorts of adult patients, from the U.S. and Italy, at different stages of disease and with different genotypes. After isolating and differentiating macrophages from PBMC, we assessed the HLA genotype and quantified the HLA-DQ2.5 mRNAs by qPCR, before and after gliadin stimulation. The results confirmed the differences in expression between DQA1*05:01 and DQB1*02:01 predisposing alleles and the non-CD associated alleles, as previously shown on other types of APCs. The gliadin challenge confirmed the differentiation of macrophages toward a proinflammatory phenotype, but above all, it triggered an increase of DQA1*05:01 mRNA, as well as a decrease of the DQB1*02:01 transcript. Furthermore, we observed a decrease in the DRB1 genes expression and a downregulation of the CIITA transactivator. In conclusion, our findings provide new evidences on the non-coordinated regulation of celiac disease DQ2.5 risk genes and support the hypothesis that gliadin could interfere in the three-dimensional arrangement of chromatin at the HLA locus.

## 1. Introduction

Celiac disease (CD) is an autoimmune enteropathy characterized by symptoms such as weight loss, chronic diarrhea, and failure to thrive. Common nonspecific symptoms include gastrointestinal manifestations, such as bloating, abdominal pain and constipation, as well as extraintestinal manifestations, such as osteoporosis, headache, iron deficiency, and chronic fatigue [1]. Celiac disease is triggered by the ingestion of gluten in genetically predisposed individuals carrying the human leukocyte antigen HLA-DQ2 or HLA-DQ8 molecules. The presence of genes coding for DQ2 and DQ8 molecules accounts for 40% of the occurrences of CD in European populations in which the frequency of the HLA-DQ2 is up by 90% and the HLA-DQ8 is between 5% and 10%. Lower frequencies of DQ2 and higher frequencies of DQ8 have also been described among CD patients in the United States (82% DQ2 and 16% DQ8). About 90% of individuals with CD carry the HLA-DQ2.5 haplotype. The risk alleles can be in *cis* configuration, located on the same chromosome (DQA1*05:01–DQB1*02:01) or in *trans* configuration, placed on the opposite chromosomes (DQA1*05:05–DQB1*03:01/DQA1*02:01–DQB1*02:02). Individuals with CD who do not express HLA-DQ2.5 usually carry either HLA-DQ2.2 (DQA1*02:01–DQB1*02:02) or HLA-DQ8 (DQA1*03–DQB1*03:02).

Antigen-presenting cells (APCs), such as dendritic cells, macrophages (MΦ), or B cells have the essential role in the pathogenesis of CD and of presenting gluten peptides to CD4^+^ naïve T cells. Antigen presentation induces differentiation and activation of CD4^+^ T lymphocytes, leading to the intestinal damage.

Blood monocytes recruited to inflamed sites differentiate into MΦ with distinct activation states: the M1 macrophage phenotypes are proinflammatory cells, whereas the M2 macrophage phenotype is indicative of cells displaying anti-inflammatory proprieties. The M1 macrophages predominantly appear at the early stage of inflammation. They release inflammatory mediators, such as IFN-γ, IL-1β, TNF-α, and IL-8, that further induce inflammation and attract leukocytes. These macrophages exhibit greater antigen-presenting ability; they are characterized by an upregulated expression of co-stimulatory molecules CD80, CD86, and CD40/CD40L, and activate T cells. During the later stages of inflammation, M2 macrophages secrete immunosuppressive cytokines, exhibit phagocytic functions, and support tissue repair, thereby helping to resolve the inflammatory process [2].

Previously published data have demonstrated that gliadin and gliadin fragments stimulated THP-1 (human monocytes cell line) cells [3] and monocytes obtained from PBMC and increased IL-8 and TNF-α production. The expression of these cytokines is more pronounced in the presence of IFN-γ. Moreover, in the presence of gliadin or IFN-γ, the monocytes HLA-DQ2^+^ released two- to three-fold more IL-8 than monocytes from HLA-DQ2^−^ donors, suggesting that this genotype predisposes monocytes to higher IL-8 secretion [4].

We have previously demonstrated the differential expression of CD-associated HLA class II risk alleles in respect to non-CD associated ones in heterozygous B-LCLs. Specifically, DQA1*05:01 and DQB1*02:01 predisposing alleles and encoding DQ2.5 molecules are expressed at high levels when carried either in *cis* (DR3 haplotype) [5] or *trans* (DR5 and DR7 haplotypes) [6] configurations. This great expression influences the strength of gliadin-specific CD4^+^ T cells response.

In previous papers, we have assessed that two mRNAs, encoding alpha and beta chains, showed a balanced expression, explained by a mechanism that ensures a coordinated gene transcription across the haplotype, and a synchronized turnover of transcripts through the binding of a protein complex with their UTRs [7]. The coordinated expression of the two mRNAs controls the density of surface heterodimers that present gliadin antigenic peptides and affect the strength of CD4^+^ T cells activation. However, the balanced expression of alpha and beta mRNAs was studied in B-LCLs in a steady state and in the absence of antigen stimulation.

In the present work, we aim to deepen our prior findings studying the HLA-DQ2.5 gene regulation in primary macrophages, from adult CD patients at different stages of disease following antigen stimulation. We confirmed the differential expression of risk genes in respect to non-CD associated alleles, but following the antigen challenge, we observed a non-coordinated variation of DQA1*05:01 and DQB1*02:01 gene expression.

## 2. Methods

### 2.1. Study Cohort

PBMCs, needed for isolating monocytes and differentiating macrophages (MΦ), were prepared from two cohorts of adult patients: a group of Italian subjects (active CD patients and CD patients in remission following the implementation of a gluten-free diet) and a group of U.S. individuals (healthy controls and CD patients on a gluten-free diet).

CD patients were considered to be in a remission state after at least 6 months of a gluten-free diet and the presentation of negative serology and intestinal mucosa recovery.

Italian subjects were enrolled at the Department of Gastroenterology of the University of Naples Federico II and the study was approved by the ethical committee of Federico II University (protocol n°178/19).

PBMCs of U.S. patients were obtained from the Massachusetts General Hospital Center for Celiac Research and Treatment Biorepository. The use of these biospecimens was approved by the MGH ethical committee for a retrospective study.

Both cohorts of subjects were genotyped for HLA genes using DQ-CD Typing Plus Kit developed by BioDiagene SRL (Palermo, Italy).

U.S. and Italian subjects selected for this study, with their genotype and diagnosis, are reported in Table 1 and Table 2.

### 2.2. PBMCs Isolation and Macrophages Differentiation

PBMCs were isolated from whole blood samples by density gradient centrifugation. Briefly, the blood samples were diluted 1:2 with phosphate buffer saline (PBS) and slowly layered on a gradient of Ficoll-Paque PLUS (GE Healthcare BioScience, Milano, Italy) solution. Once stratified, the blood was centrifuged at 400× *g* for 30 min at 22 °C and the mononuclear cells were recovered, counted, and stored at −80 °C. CD14^+^ monocytes were isolated from frozen PBMCs using CD14^+^ magnetic beads (Miltenyi Biotec, Bologna, Italy). The cell suspension was loaded onto a MACS^®^ column which, through the magnetic field of a MACS separator, retained CD14^+^ cells within the column whereas the unlabeled cells ran through. The CD14^+^ cells were eluted as positively selected cells fraction, and the efficiency of monocytes selection from PBMCs was checked by flow cytometry with a CD14-PE antibody (BD, Pharmingen™, San Diego, CA, USA). After 7 days of culture in a complete Gibco RPMI medium (Dutch modification) (Thermo Fisher, Milano, Italy) supplemented with 10% FBS (Sigma-Aldrich, Milano, Italy), 1% L-glutamine, 1% penicillin/streptomycin, 1% sodium pyruvate, and 1% nonessential amino acid solution (Gibco, Thermo Fisher, Italy), we added 50 ng/mL of M-CSF (Peprotech, London, UK). The medium was changed every 3 days, and after seven days, MΦ were differentiated from monocytes [8].

At day 7, MΦ were stimulated for 16 h with 1 mg/mL of pepsin–trypsin digested gliadin in a complete RPMI medium or with 10 µg/mL of a pool of deamidated gliadin peptides (α-gliadin 17-mer, ω-gliadin 17-mer, γ-glia DQ2-γ-I, γ-glia DQ2-γ-II, and γ-glia 26mer).

### 2.3. Cytokines Quantification

ELISA assay was performed on supernatants collected from unstimulated and gliadin-stimulated MΦ. TNF-α, IL6, and IL1β production was assessed by using a BD OptEIA Human Kit (BD Bioscience, Franklin Lakes, NJ, USA) according to the manufacturer’s instructions. Absorbance was read through a microplate reader capable of measuring absorbance at 450 nm.

### 2.4. Gene Expression Analysis

After an overnight stimulation with gliadin or with deamidated gliadin peptides, total RNA was extracted from monocytes-derived MΦ by using an RNeasy Extraction Micro Kit (Qiagen, Milano, Italy).

Reverse transcription was performed using a RevertAid First Strand cDNA Synthesis Kit (Thermo Fisher, Italy) following the manufacturer’s protocol. cDNA was used to quantify different transcripts by real-time PCR using a SYBR Green master mix (EuroClone, Pero, Italy). The allele-specific primer sequences are reported in Table 3. The relative amount of specific transcripts was calculated by the comparative cycle threshold method using an 18S transcript for normalization [9].

### 2.5. Statistical Analysis

Statistical analysis was performed on three independent experiments using the unpaired Student’s *t*-test in 2021 Microsoft Excel (16.54 version).

## 3. Results

### 3.1. Macrophages Phenotype Evaluation

Monocytes isolated from PBMCs were differentiated in vitro into macrophages, and their phenotype was assessed by measuring the production of several cytokines after gliadin stimulation. Specifically, the amount of IL-6, IL-1β, and TNF-α, three cytokines with proinflammatory function, was measured on supernatants of MΦ of healthy controls and CD patients following a GFD, prepared from a U.S. cohort. Figure 1 shows that gliadin stimulation determined an upregulation of all the cytokines analyzed.

The production of cytokines was observed in all samples independently of the genotype, and confirmed the differentiation toward a proinflammatory phenotype.

### 3.2. Analysis of Expression of HLA DQA1, DQB1, and DRB1 Genes in Macrophages

We first verified the differences in expression on MΦ between CD-associated alleles DQA1*05 and DQB1*02 and non-CD associated alleles by quantifying DQA1, DQB1, and DRB1 transcripts.

Figure 2 shows the mRNA expression of DQA1*05 respect to the DQA1*02 transcripts in MΦ from subjects of an Italian cohort carrying DR3/DR7 and DR5/DR7 genotypes (Table 1). The relative quantification showed more than 20-fold higher expression of a DQA1*05 transcript as compared to DQA1*02 mRNA in cells from both genotypes (Figure 2A). Similarly, DQB1*02 was more abundant than DQB1*03 in macrophages carrying the DR5/DR7 genotype (Figure 2B).

These results support already published data and confirm the different expression of CD predisposing alleles as compared to the non-predisposing ones [5,6].

Then, we quantified the expression of DRB1 alleles in MΦ carrying DR3/DR7 and DR5/DR7 genotypes. In the first group, the DRB1*03 allele, carried in *cis* with DQA1*05 and DQB1*02 alleles, expressed 11-fold more mRNA than the DRB1*07 allele placed on the other chromosome in linkage with DQA1*02 and DQB1*02 alleles (Figure 3A). This DRB1*07 allele transcribes a lower mRNA amount also in MΦ with the DR5/DR7 genotype, in respect to the DRB1*05 allele located in *cis* with DQA1*05 and DQB1*03. In the latter group, DRB1*05 mRNA is 8.3-fold more abundant than DRB1*07 (Figure 3B).

These results demonstrate that the high expression of DRB1*03 correlates with a high expression of DQB1*02 and DQA1*05 alleles, belonging to the same DR3 haplotype, generally associated with autoimmunity. In addition, the expression of DRB1*05 and DRB1*07 is consistent with that of the DQA1*05 and DQA1*02 alleles located, respectively, on their chromosome.

### 3.3. Antigen Stimuli Induce Variation of DQB1, DQA1 and DRB1 Transcripts Quantity

We analyzed the expression of DQ alleles after overnight stimulation of macrophages with pepsin–trypsin digested gliadin. Experiments were performed using cells from the U.S. cohorts of GFD patients and HC (Table 1) carrying DR3/DR7 and DR3/DR4 genotypes.

Figure 4 shows that gliadin stimulation induces a 13-fold increase of the DQA1*05 transcript in macrophages from both HC and GFD patients (Figure 4A). Given the fact that the genotype DR3/DR7 is homozygous for this allele, the DQB1*02 mRNA has not been analyzed.

Similarly, gliadin stimulation of DR3/DR4 MΦ induced a 9-fold increase of the DQA1*05 transcript in cells from GFD patients, and a 17-fold increase in cells from HC. Other DQ mRNAs decreases after gliadin stimulation; specifically, we measured a 2.9-fold reduction of DQA1*03, and a 5.3 and 4.3-fold decrease of DQB1*02 and DQB1*03 mRNAs, respectively, as compared to the untreated samples (Figure 4B). Slight differences in gene expression were also observed among GFD patients and HC.

DQ genes expression analysis was conducted on MΦ from Italian subjects with active CD and following a GFD, carrying DR3/DR7 and DR5/DR7 genotypes (Table 2). Figure 5A shows the results of DQ mRNA quantification in MΦ derived from GFD donors carrying the DR3/DR7 genotype. We measured a 3.3-fold increased expression of the DQA1*05 allele as well as a 6.4 and 6.6-fold decrease in DQB1*02 and DQA1*02 mRNAs, respectively, after gliadin stimulation.

Similar results were obtained from GFD patients with the DR5/DR7 genotype: the gliadin challenge induced a 5-fold increase of the DQA1*05 transcript, whereas the DQA1*02, DQB1*03, and DQB1*02 mRNAs amount was significantly decreased as compared to the unstimulated samples (4.3, 4.8, and 6.6-fold decrease, respectively, Figure 5B).

Gliadin stimulation was also performed on MΦ from patients with active CD carrying different genotypes (Table 1). Our data confirmed previous results: Figure 5C shows that the gliadin challenge triggered a significant decrease of the DQB1*02 transcript (11 fold) and a significant increase of the DQA1*05 mRNA (almost 17-fold).

The mRNA variation measured among active patients confirms data found in the MΦ of an Italian cohort of GFD patients, and it is even more pronounced.

Finally, we assessed the mRNA variation of DRB1*03 and DRB1*07 on a group of patients with the DR3/DR7 genotype, including samples from 1 patient with active CD and 3 patients following GFD (Table 1). Our data show that both DRB1*03 and DRB1*07 mRNAs decreased respectively 3.4 and 3.8-fold their amount after gliadin stimulation. (Figure 6A).

In conclusion, we observed a clear increment of DQA1*05 mRNA associated with CD, and a decrease of DQB1 and DRB1 mRNAs expression in all the subjects tested, independently of their genotype, although the extent of variations was different.

To verify whether the uncoordinated variation of DQB1*02 and DQA1*05 mRNAs was related to the processing, we analyzed the HLA-DQ expression after stimulation of macrophages with a pool of gliadin peptides (α-gliadin 17-mer, ω-gliadin 17-mer, γ-glia DQ2-γ-I, γ-glia DQ2-γ-II, and γ-glia 26mer). Stimulation was performed on the MΦ of GFD patients with the DR5/DR7 genotype. We observed 3.5-fold increase of the DQA1*05 transcript (Figure 7), a 4.5-fold decrease of the DQB1*02 amount (Figure 7), and 3.3 and 3-fold diminution of DRB1*05 and DRB1*07 mRNA, respectively (Figure 6B).

The fold variation of CD-related mRNAs measured after stimulation with gliadin peptides indicates that it is not correlated to the processing of antigenic protein.

We verified whether the DQB1*02 decrease affected the surface expression after gliadin stimulation by flow cytometry. We performed the experiment on two samples of MΦ from homozygous GFD patients, and we measured the MFI of the HLA-DQ2 expression. We observed that, as expected, the decrease of DQB1*02 mRNA induced a 25% reduction of the total surface HLA-DQ (data not shown).

These findings demonstrated that the regulation of CD risk alleles DQA1*05 and DQB1*02 in macrophages was not coordinated. The antigen challenge, either gliadin or peptides, determined a decrease of DQB1*02 and an increase of DQA1*05, with a variable extent and influenced the surface DQ molecule expression.

### 3.4. Gliadin Stimulation Induces CIITA Downregulation

To unravel the mechanism regulating the non-coordinated expression of DQA1*05 and DQB1*02 alleles following gliadin stimulation of MΦ, we measured the amount of the CIITA master transactivator by qPCR. Figure 8 shows a clear decrease of mRNA expression, ranging from 7 to 20-fold, in all samples analyzed from an Italian cohort, independently of their genotypes.

This result explains the downregulation of transcription at the HLA class II locus.

## 4. Discussion

In this work, we analyzed for the first time the expression of DQA1*05:01 and DQB1*02:01 alleles in primary macrophages, obtained by peripheral blood of adult celiac patients at diagnosis and in remission while following a GFD. The study was conducted on a cohort of U.S. patients, mainly with the DR3/DR7 and DR3/DR4 genotypes, and on a cohort of Italian patients mostly carrying DR3/DR7 and DR5/DR7 genotypes. Our data confirmed the proinflammatory phenotype of MΦ samples that secrete IL-6, IL-1β, and TNF-α triggered by gliadin stimulation, independently of diagnosis and genotypes.

We established the different expression of DQA1*05:01 and DQB1*02:01 risk alleles as compared to non-CD associated alleles in macrophages, as already demonstrated in B-LCLs. Moreover, we measured the DRB1 gene expression in samples carrying the DR3/DR7 genotypes and we observed for the first time differences in expression of the DRB1*03 allele in linkage with DQA1*05:01 and DQB1*02:01, belonging to DR3 haplotype, as compared to DRB1*07 allele on the other chromosome. Similarly, the DRB1*07 allele shows a lower expression than the DRB1*05 allele in DR5/DR7 cells. These results suggest that the regulation of the DRB1 gene depends on the haplotypic contest in which it is located.

Following overnight gliadin stimulation of MΦ, we detected a significant increase of the CD-associated DQA1*05 mRNA and a decrease of the CD-associated DQB1*02 transcript. The DQA1*05 mRNA upregulation and DQB1*02 downregulation following antigen stimulation, although of a different extent, was observed in all MΦ samples analyzed, independently of their geographical origin (Italy or USA), their genotype, and their clinical status (active CD patients, donors in remission following a GFD and HC). Following stimulation of MΦ with gliadin protein or peptides, all DRB1 alleles are downregulated.

Moreover, we demonstrated that the decrease of mRNAs is associated with a reduction of CIITA expression after gliadin stimulation. This result is not surprising because our measurements follow the antigen presentation when HLA genes have achieved their function. More surprising, instead, is the upregulation of the DQA1*05 allele despite the CIITA decrease. We propose that the mechanism of the non-coordinated regulation of two alleles is probably due to the three-dimensional structure of chromatin.

The HLA class II genes transcription is regulated by a complex network of DNA-binding factors and conserved cis-acting WXY-boxes in their promoters, under the control of the master regulator CIITA [10]. This complex forms an enhanceosome that recruits chromatin modifiers and the general transcription machinery, leading to the expression of these genes.

Many other regulatory elements, defined X boxes, were mapped across the HLA class II locus corresponding to peaks of histone acetylation [11]. Among these sites, the best characterized is XL9 [12], located in the intergenic region between HLA-DRB1 and HLA-DQA1 genes, that interacts with the insulator CCCTC binding factor (CTCF). CTCF has been shown to define and insulate regions of regulatory activity within the genome by functioning as an enhancer blocker, or by preventing the spread of heterochromatin into active genes [13]. CTCF works together with the cohesin complex to form long-range chromatin loops between proper binding sites, via a mechanism involving loop extrusion.

Twelve CTCF sites have been mapped across the HLA class II locus. They have the general function to organize chromatin into large regulatory loops that limit the effects of enhancers. All interactions of CTCF with the MHC-II gene promoter regions occurred only in the CIITA-expressing cell line, suggesting that the two factors interact directly or indirectly through a complex. In addition, each promoter cooperates with multiple CTCF sites and these cooperate among them. In general, the highest frequency of interaction was observed with the nearest upstream or downstream neighbor. The strength of interactions and the function of the CTCF factor depends on the spatial proximity/distance and the polarity of sites. In fact, the convergent orientation between the two CTCF sites is much more frequent in the genome and absolutely preferred for the correct binding of cohesin protein [14] and proper enhancer–promoter interactions.

The CTCF site, located upstream DRA gene (defined C1), interacts at a high frequency with its promoter and, to a lesser degree, with XL9, which is 209 kb away. Interactions of HLA-DRB1 and HLA-DQA1 with XL9 were previously reported [15]. In fact, although HLA-DRB1 interacted at the same frequency with two flanking C1 and XL9 sites, HLA-DQA1 and HLA-DQB1 showed the highest-frequency interactions with XL9 and C2.

Recently, a new super enhancer named DR/DQ-SE has been identified between DRB1 and DQA1, and is required for the optimal expression of the DRB1, DQA1, and DQB1 genes, favoring the interactions between promoters, XL9 and CTCF binding sites within the locus. DR/DQ-SE function is associated with active chromatin modifications, consisting in the enrichment of H3K27ac, H3K4me1, and H3K4me3 factors. Its deletion resulted, instead, in a lower recruitment of CIITA to the promoters and a decrease of histone modifications associated with active chromatin [16].

Our results on the gliadin-stimulated macrophages show similarities with findings obtained by the analysis of monocytes from septic patients with immune suppression. In these patients, the strong inflammation determines a downregulation of some HLA class II molecules and a concomitant decrease of CIITA expression, caused by modifications of histone acetylation profiles. Specifically, sepsis was associated with a selective increase of CTCF enrichment at three binding sites corresponding to C1, XL9, and C2, whereas no differential CTCF occupancy was detected at the other binding sites across the locus [17,18].

We propose that the downregulation of MHC-II and CIITA expression by gliadin stimulated macrophages might be explained by a decrease of histone acetylation at CTCF sites surrounding DRB1, DQA1, and DQB1 genes. The three-dimensional arrangement of chromatin might place the DR-DQ/SE very close to the DQA1*05 promoter, that, irrespective of XL9 enhancer-blocking function, sustains the high expression of this allele after antigen stimulation.

It is intriguing to speculate that either the enhancer DR/DQ-SE and both insulators XL9 and CTCF binding sites orchestrate multiple interactions across the region involving CIITA, CTCF, cohesion, and transcriptional complex, and thereby modulate the levels of MHC-II gene expression in different APCs.

Our previous papers [6,19] and other previously published data [20,21] widely demonstrated the differential expression of HLA haplotypes associated with autoimmune diseases. Many polymorphisms are identified in the intergenic regions that affect the binding of transcription factors or the levels of histone H3K27 acetylation. Moreover, enhancer histone-QTLs are enriched on autoimmune disease risk haplotypes [22], confirming that SNPs may function as epigenetic modulators of gene expression.

In conclusion, the gliadin challenge affects both functions of macrophages, the inflammatory response and the antigen presentation, the latter regulated by HLA-DQ2.5 risk alleles expression.

## Figures and Tables

**Figure 1 biomedicines-10-00063-f001:**
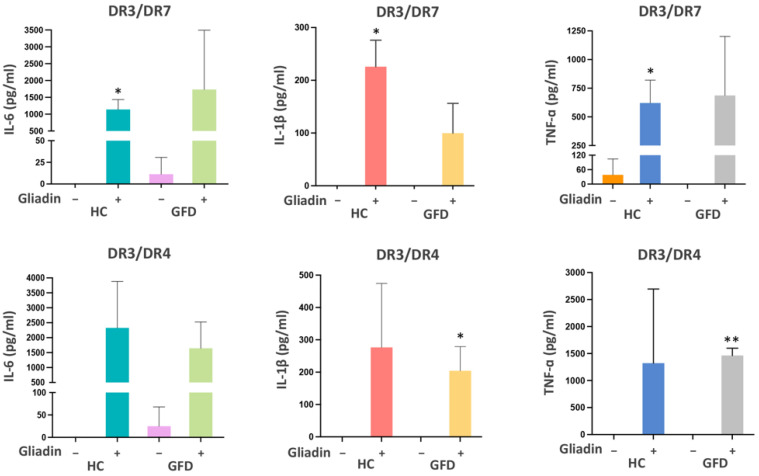
Proinflammatory cytokines production after gliadin stimulation. Quantification of proinflammatory cytokines on supernatants of macrophages after overnight gliadin stimulation in healthy controls (HC) (*n* = 3: FFL340, FFL665, and FFL329) and CD patients following a gluten-free diet (GFD) *(n* = 3: CHM144A, CHM150A, and CHM019A) carrying DR3/DR7 genotype and in HC (*n* = 3: FFL605, FFL642, and FFL1119) and GFD patients (*n* = 3: CHM13C, CHM88A, and FFL639) with DR3/DR4 genotype (* *p* < 0.05, ** *p* < 0.01).

**Figure 2 biomedicines-10-00063-f002:**
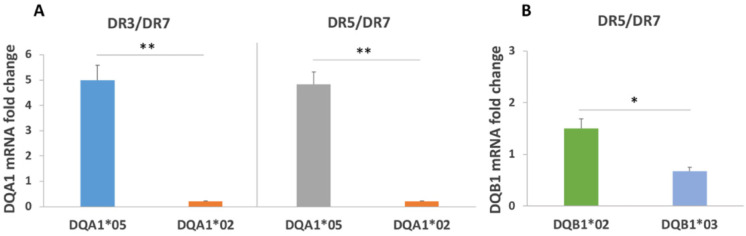
Expression of DQA1 and DQB1 genes. (**A**) The DQA1*05 mRNA shows a 25-fold variation in respect to DQA1*02 mRNA in DR3/DR7 MΦ (*n* = 4: CD20, CD25, CD31, and CD35) and a 24-fold variation in the DR5/DR7 genotype. (*n* = 2: CD26 and CD34). (**B**) DQB1*02 mRNA is also more abundant than the DQB1*03 mRNA in MΦ with the DR5/DR7 genotype (*n* = 2: CD26 and CD34). (* *p* < 0.05, ** *p* < 0.01).

**Figure 3 biomedicines-10-00063-f003:**
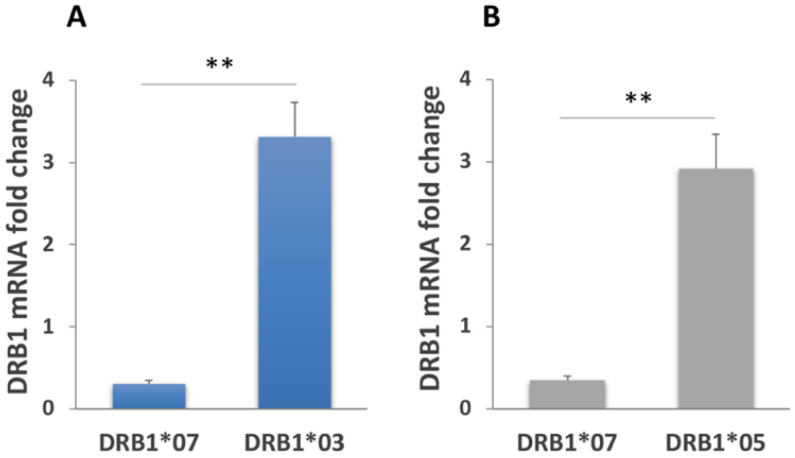
Expression of DRB1 alleles. (**A**) DRB1*03 mRNA is 11-fold more expressed than the DRB1*07 transcript in macrophages carrying the DR3/DR7 genotype (*n* = 4: CD20, CD25, CD31, and CD35). (**B**) DRB1*05 mRNA is 8.3-fold more abundant than the DRB1*07 mRNA in MΦ with the DR5/DR7 genotype (*n* = 5: CD07, CD11, CD21, CD26, and CD55). (** *p* < 0.01).

**Figure 4 biomedicines-10-00063-f004:**
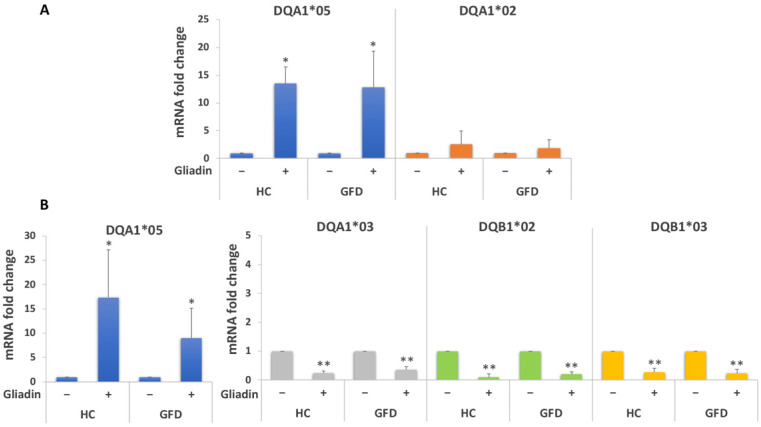
Analysis of DQ genes expression of macrophages from a U.S. cohort stimulated with gliadin. (**A**) DQA1*05 and DQA1*02 mRNAs fold variation in HC (*n* = 3: FFL340, FFL665, and FFL329) and GFD patients (*n* = 3: CHM144A, CHM150A, and CHM019A) with the DR3/DR7 genotype after overnight gliadin stimulation (* *p* < 0.05). (**B**) DQA1*05, DQA1*03, DQB1*02, and DQB1*03 mRNAs fold variation in HC (*n* = 3: FFL605, FFL642, and FFL1119) and GFD patients (*n* = 3: CHM13C, CHM88A, and FFL639) carrying the DR3/DR4 genotype after overnight gliadin stimulation (* *p* < 0.05, ** *p* < 0.01).

**Figure 5 biomedicines-10-00063-f005:**
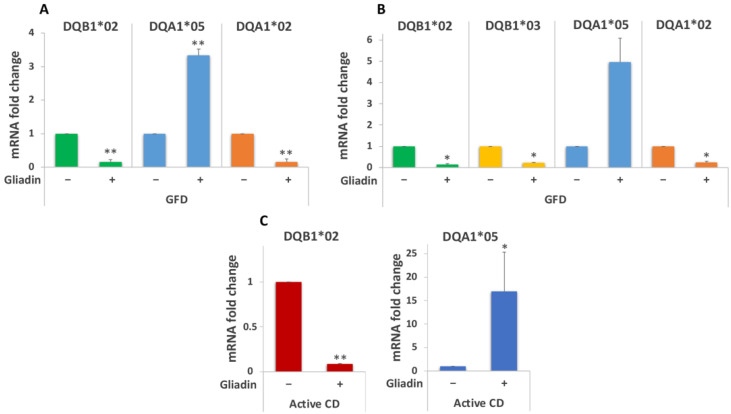
Analysis of DQ genes expression of macrophages from an Italian cohort. (**A**) DQB1*02, DQA1*05, and DQA1*02 mRNAs fold variation in GFD patients with the DR3/DR7 genotype (*n* = 3: CD20, CD25, and CD31) after overnight gliadin stimulation (** *p* < 0.01). (**B**) DQB1*02, DQB1*03, DQA1*05, and DQA1*02 mRNAs fold variation in GFD patients with the DR5/DR7 genotype (*n* = 2: CD26 and CD30) after overnight gliadin stimulation (* *p* < 0.05). (**C**) DQB1*02 and DQA1*05 mRNAs fold variation in active CD patients (*n* = 3: CD14, CD34, and CD35) after overnight gliadin stimulation (* *p* < 0.05, ** *p* < 0.01).

**Figure 6 biomedicines-10-00063-f006:**
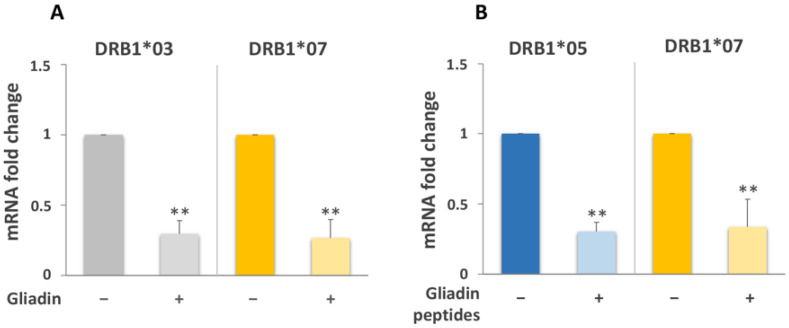
Expression of DRB1 alleles. (**A**) Gliadin stimulation induced a 3.4 and 3.8-fold decrease of DRB1*03 and DRB1*07 mRNAs, respectively, in the MΦ of CD patients with the DR3/DR7 genotype (*n* = 4: CD20, CD25, CD31, and CD35). (**B**) MΦ of DR5/DR7 patients showed a 3.3 and 3-fold decrease of DRB1*05 and DRB1*07 mRNAs, respectively, after gliadin peptides stimulation (*n* = 5: CD07, CD11, CD21, CD26, and CD55). (** *p* < 0.01).

**Figure 7 biomedicines-10-00063-f007:**
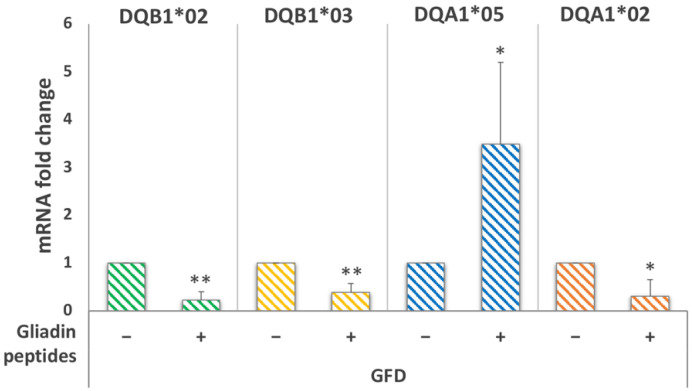
Analysis of DQB1 and DQA1 gene expression after stimulation of MΦ with a pool of gliadin peptides. DQB1*02, DQB1*03, DQA1*05, and DQA1*02 mRNAs fold variation in GFD patients (*n* = 5: CD07, CD21, CD24, CD26, and CD55) with the DR5/DR7 genotype after overnight stimulation with a pool of gliadin peptides (* *p* < 0.05, ** *p* < 0.01).

**Figure 8 biomedicines-10-00063-f008:**
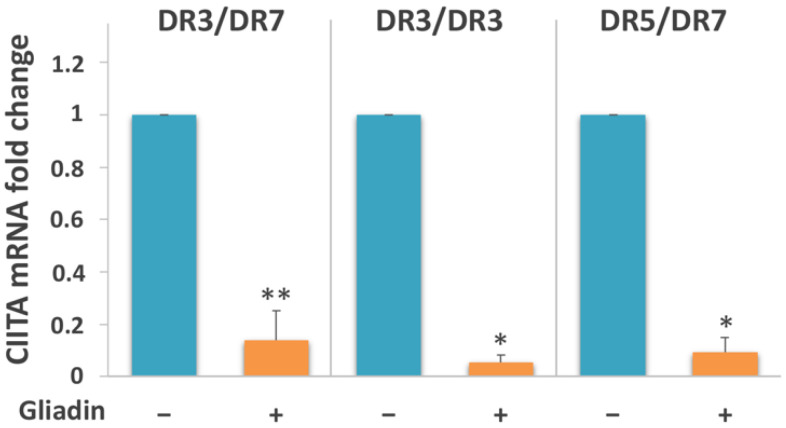
Analysis of expression of CIITA mRNA. CIITA mRNA decreased 7-fold in MΦ carrying the DR3/DR7 genotype (*n* = 4: CD20, CD25, CD31, and CD35), 20-fold in samples with the DR3/DR3 genotype (*n* = 2: CD14 and CD18) and 11-fold in MΦ carrying the DR5/DR7 genotype (*n* = 2: CD26 and CD34) after overnight gliadin stimulation (* *p* < 0.05, ** *p* < 0.01).

**Table 1 biomedicines-10-00063-t001:** HLA-DQ and -DR genotypes of U.S. celiac disease patients following a gluten-free diet (CD on GFD) and healthy controls (HC) chosen for macrophages analysis.

ID Sample	DQ Genotype	DR Genotype	Diagnosis
FFL 340	DQ2/DQ2	DR3/DR7	Healthy control
FFL 665	DQ2/DQ2	DR3/DR7	Healthy control
FFL 329	DQ2/DQ2	DR3/DR7	Healthy control
CHM 144A	DQ2/DQ2	DR3/DR7	CD on GFD
CHM 150A	DQ2/DQ2	DR3/DR7	CD on GFD
CHM 019A	DQ2/DQ2	DR3/DR7	CD on GFD
FFL 605	DQ2/DQ8	DR3/DR4	Healthy control
FFL 642	DQ2/DQ8	DR3/DR4	Healthy control
FFL 1119	DQ2/DQ8	DR3/DR4	Healthy control
CHM 13C	DQ2/DQ8	DR3/DR4	CD on GFD
CHM 88A	DQ2/DQ8	DR3/DR4	CD on GFD
FFL 639	DQ2/DQ8	DR3/DR4	CD on GFD

**Table 2 biomedicines-10-00063-t002:** HLA-DQ and -DR genotypes of Italian celiac disease patients chosen for macrophages analysis.

ID Sample	DQ Genotype	DR Genotype	Diagnosis
CD14	DQ2/DQ2	DR3/DR3	active CD
CD18	DQ2/DQ2	DR3/DR3	CD on GFD
CD20	DQ2/DQ2	DR3/DR7	CD on GFD
CD25	DQ2/DQ2	DR3/DR7	CD on GFD
CD31	DQ2/DQ2	DR3/DR7	CD on GFD
CD35	DQ2/DQ2	DR3/DR7	active CD
CD07	DQ2/DQ7	DR5/DR7	CD on GFD
CD11	DQ2/DQ7	DR5/DR7	CD on GFD
CD21	DQ2/DQ7	DR5/DR7	CD on GFD
CD24	DQ2/DQ7	DR5/DR7	CD on GFD
CD26	DQ2/DQ7	DR5/DR7	CD on GFD
CD30	DQ2/DQ7	DR5/DR7	CD on GFD
CD34	DQ2/DQ7	DR5/DR7	active CD
CD55	DQ2/DQ7	DR5/DR7	CD on GFD

**Table 3 biomedicines-10-00063-t003:** Primers used for qPCR.

Gene	Primer	Sequence
GAPDH	GAPDH-F	GAAGGTGAAGGTCGGAGTC
	GAPDH-R	GAAGATGGTGATGGGATTTC
β-ACTIN	ACTa-F	TCATGAAGTGTGACGTTGACA
	ACTa-R	CCTAGAAGCATTTGCGGTGCAC
18S	18S-FW	AGAAACGGCTACCACATCCA
	18S-RW	CCCTCCAATGGATCCTCGTT
HLA-DQA1*05	DQA1*05-F	TGGTGTTTGCCTGTTCTCAGAC
	DQA1*R	GGAGACTTGGAAAACACTGTGACC
HLA-DQA1*02	DQA1*02-F	AAGTTGCCTCTGTTCCACAGAC
	DQA1*R	GGAGACTTGGAAAACACTGTGACC
HLA-DQA1*03	DQA1*03-F	CTCTGTTCCGCAGATTTAGAAGA
	DQA1*R	GGAGACTTGGAAAACACTGTGACC
HLA-DQB1*02	DQB1*02-F	TCTTGTGAGCAGAAGCATCT
	DQB1*R	CAGGATCTGGAAGGTCCAGT
HLA-DQB1*03	DQB1*03-F	CGGAGTTGGACACGGTGTGC
	DQB1*R	CAGGATCTGGAAGGTCCAGT
CIITA	C2T-F	CCGACACAGACACCATCAAC
	C2T-R	CTTTTCTGCCCAACTTCTGC
HLA-DRB1*03	DR3-F	CTCCTGGAGCAGAAGCGGGG
	DR3-R	CCGGAACCACCTGACTTCAAT
HLA-DRB1*07	DR7-F	CTGTGGCAGGGTAAGTATAAGTG
	DR7-R	GCCTGGATAGAAACCACTCAC
HLA-DRB1*05	DR5-F	TTGGAGTACTCTACGTCTGAGTG
	DR7-R	GCCTGGATAGAAACCACTCAC

## Data Availability

Data sharing was not applicable to this article as no datasets were generated or analyzed during the current study.

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
