# Peer review of "Effect of Gliadin Stimulation on HLA-DQ2.5 Gene Expression in Macrophages from Adult Celiac Disease Patients"

_biomedicines, 2021, doi:10.3390/biomedicines10010063_

Round 1
Reviewer 1 Report
The research article is well-explained and served the purpose of the study. In this work, the expression of Celiac Disease-associated HLA-DQ2.5 risk alleles on macrophages isolated by two cohorts of adult CD patients, from the U.S. and Italy, at a different stage of disease and with different genotypes was studied.
Data is well-presented; however, the manuscript needs some minor changes before acceptance that are described below:
- Statistical application is missing in Figure1.
- Results are concluded at the end of each figure. It will be better to conclude the overall result in the end as well.
- Some minor grammar mistakes need to be fixed.
Author Response
Statistical analysis has been added to fig 1
The conclusion of each experiments has been included at the end of result paragraphs
The English editing has been performed
Reviewer 2 Report
A very interesting study exploring, the expression of DQA1*05:01 and
DQB1*02:01 alleles in primary macrophages, obtained by peripheral blood of adult celiac patients at the moment of diagnosis and in remission, during a gluten-free diet. Only minor queries before acceptance:
In the statistical analysis subchapter, You should specify the program you used to perform the statistical analysis, its maker, version, and location.
In the introduction, celiac disease symptoms should be assessed; you could add a sentence such as: "The symptoms classically include weight loss, chronic diarrhea, and failure to thrive. Non-specific
symptoms are more common and include gastrointestinal manifestations, such as bloating, abdominal
pain, constipation, as well as extra-intestinal manifestations, as osteoporosis, headache, iron deficiency,
and chronic fatigue" and a citation such as: https://doi.org/10.3390/medicina55090578
Good Luck!
Author Response
The program used for statistical analysis has been specified
The sentence describing general celiac symptoms has been added in the introduction